# Developing and Validating an Instrument for Measuring Teachers' Informatization Teaching Ability in Primary and Secondary Schools in China for the Sustainable Development of Education Informatization

Suping Yi [1], Rustam Shadiev [1,*], Ruwei Yun [1] and Yefeng Lu [2]

[1] School of Education Science, Nanjing Normal University, Nanjing 210097, China; yisupinger@gmail.com (S.Y.); yunruwei@njnu.edu.cn (R.Y.)

[2] College of Education, Zhejiang University, Hangzhou 310028, China; 12103004@zju.edu.cn

* Correspondence: rustamsh@gmail.com

**Abstract:** Teachers' informatization teaching ability (TITA) is the core ability of teachers to engage in educational activities in the informatization environment. It is an essential indicator of the quality of education and affects teachers' professional development in the technological age. To get a precise teachers' informatization teaching ability scale and to measure TITA more accurately, the quality of existing scales needs to be improved. This study analyzed and generalized definitions, models and frameworks for TITA, proposed a four-dimension model (teachers' informatization teaching metacognitive ability, informatization teaching design ability, informatization teaching implementation ability and informatization teaching evaluation ability), and developed the TITA scale. Three experts were involved in the content validity of the TITA scale process. A total of 403 valid questionnaires answered by Chinese primary and secondary school teachers were used to test the reliability and convergent validity of the scale. The results showed that the TITA scale has high reliability and good validity, and it can be used to evaluate TITA in future studies. The TITA scale also provided a theoretical framework to help teachers consider how to transfer knowledge and skills to students by various technologies more effectively.

**Keywords:** teachers' informatization teaching ability; instrument development and validation; professional development; primary and secondary school teachers

## 1. Introduction

Sustainability refers to the ability to support a process continuously over time in the broadest sense [1]. In a narrow sense, such as in education, sustainability seeks to keep education consistent with social development [2,3]. In current society, education for sustainable development has received increasing attention. With the advent of the information age, the in-depth development of sustainable education requires extensive and flexible application of information technology. Emerging technologies are changing traditional teaching models at a fantastic rate, such as teaching time and space, teaching methods, etc., which have been significantly impacted. Instead, diversified resources, vivid content presentation forms and freedom of learning time and space have been replaced. While artificial intelligence (AI), cloud computing, virtual reality (AR) and other emerging technologies bring dividends to education, technology has also impacted education, especially teachers, who play a leading role in the teaching process. In current education, it is necessary for teachers to use various modern technologies to efficiently impart knowledge and skills needed in the 21st century to students. From this context, teachers' informatization teaching ability (TITA) is gradually being noticed by educators and educators. The development of TITA is not only a vital force to promote the sustainable

development of education but also a pivotal link to promoting the sustainable development of education informatization.

TITA plays a crucial role in contributing to teachers' professional development in a technological society, and TITA can be seen as a booster that promotes a virtuous circle in the informatization teaching environment. That is, teachers' degree of informatization teaching ability determines the release limit of technological benefits in education. Recently, TITA has gained increasing attention as an essential aspect of teachers' application ability of information technology (TAAIT) in China [4–6]. Researchers and educators believe that the promotion of TITA can cultivate the practical wisdom of teacher automation that leads to facilitating teaching and learning [7,8].

TITA has been extensively studied by many scholars. For example, existing studies on TITA already have various definitions, models or frameworks. However, to the best of our knowledge, not all factors were explained clearly in the existing literature. That is, empirical research does not thoroughly examine all existing factors, and some still do not have a solid theoretical foundation [9,10]. Furthermore, some studies do not ensure the quality of measurement instruments [11,12].

Therefore, the priority was to develop an instrument for measuring TITA from the perspective of theoretical deductions so that they could be more targeted for supporting teachers' teaching and students' learning. The following research questions guided the present study:

1. How is TITA defined in the existing literature?
2. What is the theoretical framework of TITA?
3. What are the reliability and convergent validity of TITA?

For the first question, this study defined the TITA by literature review. Then this study constructed a theoretical framework for the TITA scale based on a literature review and empirical research for the second question. The TITA scale was tested and validated by samples collected from Chinese primary and secondary school teachers using exploratory factor analysis (EFA), reliability analysis and confirmatory factor analysis (CFA) to answer the third question.

## 2. Literature Review

### 2.1. Context of Developing Teachers' Informatization Teaching Ability (TITA)

The development and improvement of TITA have always been the "last mile" of education informatization, and it is the key to the in-depth integration of information technology and education and teaching. Paying attention to TITA is the professional aspiration of in-service teachers and the essential requirement to improve the quality of education in the whole society.

From the point of view of national policies, in China, the Ministry of Education promulgated the Education Informatization 2.0 Action Plan in 2018, which pointed out that "it is necessary to promote teachers to actively adapt to new technological changes such as informatization and artificial intelligence, and to strengthen the cultivation of teachers' information literacy and ITA" [5]. In 2019, the Ministry of Education clearly stated in the Opinions on the Implementation of the National Primary and Middle School Teachers' Information Technology Application Ability Enhancement Project 2.0 that "teachers should be encouraged to use online learning spaces, teacher workshops, research communities, etc., and integrate online resources with offline seminars to improve TITA".

From teachers' perspective, the emergence of blended learning, online teaching, etc., requires teachers to have excellent literacy in an informatization environment. Among them, the improvement of ITA is the biggest challenge for teachers. Take the large-scale online education during the COVID-19 period as an example. Gao and Zhang's [13] study reported that the teachers hope to get the help of modern information technology knowledge and skills during online teaching. In addition, some teachers are anxious about online education and feel that it is difficult for them to adapt to online teaching activities in a short time. Fu and Zhou [14] found in the survey that most teachers have a strong demand

for online teaching technical guidance during the COVID-19 period. In a recent survey, researchers found that teachers still show continuous enthusiasm for applying information technology in the teaching process despite the restoration of offline teaching [15]. Some teachers who have not used information technology before are now trying to use simple technology for teaching activities.

The development of a TITA scale suitable for the education informatization 2.0 era not only meets the needs of China's policies but also provides support for the scientific evaluation of TITA. Therefore, it is meaningful in this context to develop the TITA scale in order to contribute to the promotion and development of education informatization in China.

### 2.2. Definition of Teachers' Informatization Teaching Ability (TITA)

A summary of definitions of TITA has been proposed in the literature. Researchers and educators generally gave definitions from three aspects. The first aspect highlights the value orientation of TITA. Du [16] suggests that TITA is the comprehensive ability of teachers to use information resources to carry out instructional activities and complete teaching tasks to promote students' development. The second aspect emphasizes the application of technology in the teaching process. Qi [17] considers TITA as the ability of teachers to use modern educational technology and equipment to educate students and solve problems. The last aspect focuses on the integration of technology and the teaching process. Liu et al. [9] suggest that TITA refers to teachers using information technology to arrange all links and elements of the teaching process scientifically.

Although the definitions of TITA are very broad, a consensus of the conceptualizations of the term has somehow been reached, which constitutes the definition of TITA in this study. TITA is a collection of abilities, which is composed of several informatization teaching sub-abilities; TITA is a teacher's professional ability developed in the process of informatization teaching practice; TITA is applied in a certain informatization teaching situation [18]. This concept first acknowledges that TITA's core ability is undertaking complex teaching tasks by using educational technologies in the classroom. Meanwhile, according to the division of the teaching process proposed by Jackson [19] and Clark and Peterson [20], TITA is limited to periods that cover time during the lessons and is extended to time before and after the lessons. The ultimate goals of TITA are to lead students' learning and help teachers' professional development [8,21]. Based on the above content, TITA in this study is defined as follows: It is a comprehensive practical competence that teachers use modern technologies to integrate and innovate teaching design, implementation, evaluation and other links of the teaching process.

### 2.3. Comparison of Teachers' Informatization Teaching Ability (TITA) and TPACK

There are plenty of measurement instruments that focus on teachers' competence in the process of teaching in the technology environment. TPACK is the most famous among these extant instruments and is generally applied. TPACK describes the interaction between technology knowledge, pedagogy knowledge and content knowledge [22]. When using TPACK for instructional design, teachers need to comprehensively consider their knowledge levels in three aspects and integrate them into the same teaching activity. For example, in an English-speaking practice activity, teachers need to consider the knowledge of oral English and logical thinking (content), the knowledge of instructional design, classroom organization (pedagogy) and the knowledge of how to use technology to make the activity more efficient (technology). Therefore, technology is not the main focus but is considered by teachers in parallel with the pedagogy and content in TPACK.

In this study, we develop a TITA instrument based on the teaching process in class, including before class, during class and after class. TITA takes "technology" as the central perspective to consider how to help the integration of technology and teaching effectively. When teachers use technology, they need to think about such questions: to what extent do I need to integrate technology with teaching? At this link, what does technology allow me

to do? In what ways does technology allow me to incorporate it into teaching activities? These are issues we need to consider from the perspective of technology. Thus, teachers can use TITA to examine how well they integrate technology into their teaching.

Combined with TPACK-related instruments, this study designed a TITA measurement instrument to help researchers and scholars to pay attention to how to integrate existing technologies into the teaching process in the informatization class.

*2.4. Measures of Teachers' Informatization Teaching Ability (TITA)*

There are two official standards related to TITA in China. In 2004, the Ministry of Education promulgated the Standard of Teacher's Ability of Applying Educational Technology (TAAET) in Primary and Secondary Schools [23]. In 2014, the Ministry of Education issued the Standard of Teacher's Ability of Applying Information Technology (TAAIT) in Primary and Secondary Schools [24]. Both are influential standards and are widely used in empirical investigations by researchers and educators [25,26]. However, it needs to be admitted that TAAET and TAAIT are different from TITA. TITA gives close attention to the innovation of instruction and technologies compared to TAAET and TAAIT. Moreover, with frequent innovations and developments related to instruction and integration of technology to support it, the framework of TITA needs to be updated from time to time considering changes in actual teaching situations.

Although researchers have shown an increased interest in TITA, very few studies have explored existing instruments to measure TITA. The survey by Liu et al. [27] focused on preservice teachers' classroom teaching and measured TITA from four dimensions: teaching awareness, teaching knowledge, teaching skills and high-level capabilities. Su et al. [28] adopted a fuzzy comprehensive evaluation method to assess TITA, including professional foundation, teaching design ability, teaching implementation and monitoring ability, teaching evaluation ability and teaching research ability. Liu [29] designed four items of TITA through the Delphi method: teaching professional basis, teaching design ability, teaching implementation and monitoring ability, teaching evaluation ability, and teaching research ability. In addition, Mao et al. [30] applied the TITA scale to their study to discuss the relationships between the university evaluation system, work pressure and TITA. In a study conducted by Guan and Huang [31], scholars investigated the current development situation of college teachers' information technology teaching ability based on the TITA framework. Additionally, several studies explored the factors restricting the development of TITA based on their TITA models [9,32].

In spite of the fact that articles on measurement instruments of TITA are quite new, most scholars did not ensure their quality. For example, most existing instruments have suffered from a lack of instrumental sensitivity, especially when researchers report the results of reliability and validity, exploratory factor analysis (EFA), confirmatory factor analysis (CFA) and weight calculation [9,27–30,32].

In order to improve the quality of the TITA instruments, this study provides a new tool according to the teaching process to assess TITA. Furthermore, the TITA scale's quality is demonstrated by reliability analysis, CFA and EFA that we carried out in the present study.

## 3. Theoretical Framework for Instrument Development

*3.1. The Theoretical Framework of TITA*

As Borich [32] said, "teaching is complex and difficult, requiring unique abilities and structures." In the context of informatization, there is still no consensus on what kind of teaching ability teachers need. Teaching ability is the ability demonstrated in the class. Therefore, it is a more effective analytical paradigm to explore the components of the TITA measurement instrument from the perspective of the classroom teaching process.

Jackson [19], an American educator, divides teaching into two stages, "before action" and "in action". The former refers to the stage of lesson preparation, selection of teaching methods and teaching materials before teaching, and the latter refers to the stage of interaction between teachers and students in the classroom. Afterward, Clark and Peterson [20]

proposed a third stage, the "post-action" stage, which refers to how teachers decide on the following teaching stage after the lesson. Therefore, from the sequence of events, the teaching process can be divided into three stages: before the teaching action, during the teaching action and after the teaching action.

Based on the interpretation of TAAIT and TPACK in a deep sense, this study summarizes the relevant elements of teachers' informatization teaching ability structure and comprehensively considers the development of actual classroom events. The TITA framework comprises four dimensions: teachers' informatization teaching metacognitive ability (TITMA), teachers' informatization teaching design ability (TITDA), teachers' informatization teaching implementation ability (TITIA) and teachers' informatization teaching evaluation ability (TITEA). All dimensions can be seen in Table 1.

**Table 1.** First-level dimensions and source.

| Dimension | Stage | Source |
|---|---|---|
| Teachers' informatization teaching metacognitive ability (TITMA) | Before lesson; after lesson | TPACK; TAAIT; relevant models |
| Teachers' informatization teaching design ability (TITDA) | Before lesson | TAAIT; relevant models |
| Teachers' informatization teaching implementation ability (TITIA) | During lesson | TAAIT; relevant models |
| Teachers' informatization teaching evaluation ability (TITEA) | During lesson; after lesson; before lesson | TAAIT; relevant models |

### 3.2. Instrument Development for Assessing TITA

What dimensions does TITA include, or which domains of educational practices reveal TITA? These issues have received considerable attention in the existing literature. This study summarized the classifications of some influential research models. We determined the second-level and third-level measurement items based on TPACK, TAAIT and relevant models.

The first dimension is teachers' informatization teaching metacognitive ability (TITMA), and it relates to teachers' knowledge and understanding of applying educational technologies to the teaching process. To some extent, the level of TITMA determines the effects of integrating technology and teaching. Therefore, TITMA performs the following tasks in a technology-rich teaching environment: update teaching concepts; learn new teaching knowledge and skills; and raise self-development awareness [33–36]. From the statements and points of view in Dimension 1: TITMA, we complied 5 second-level items and 12 third-level measurement items, as shown in Table 2.

**Table 2.** Second and third-level items of TITMA and resource.

| Second-Level Items | Third-Level Measurement Items | Resource |
|---|---|---|
| Informatization teaching philosophy | A1: TITA is a core component of teachers' competence. | Xu and Chen [37] |
| | A2: Technology is important for teaching. | Schmid et al. [38] |
| | A3: It is necessary to use technology to optimize teaching. | Su et al. [39] |
| Informatization teaching skills | A4: I explore technological resources for students' learning. | Bilici et al. [40] |
| | A5: I grasp how to use different multimedia tools. | Schmid et al. [38] |
| | A6: I have several methods to get digital instructional resources. | Bilici et al. [40] |

| Second-Level Items | Third-Level Measurement Items | Resource |
|---|---|---|
| Informatization pedagogical knowledge | A7: I master comprehensive knowledge of education. | Schmid et al. [38] |
| | A8: I have pedagogical knowledge. | Schmid et al. [38] |
| | A9: I possess computer ethics and information security knowledge. | Wen and Shih [41] |
| Informatization professional development | A10: I use online learning communities to improve myself. | Hsu [42] |
| | A11: I communicate with experts by technological resources to improve myself. | Wu [23] |
| | A12: I solve teaching problems with common software tools. | Sun et al. [43] |

The second dimension is teachers' informatization teaching design ability (TITDA) which occurs before the lessons. Instructional design is the blueprint for teachers' teaching. Several educators point out that to be in possession of TITDA, teachers should design activities supported by diverse technologies; consider students' learning styles; formulate actual instructional objectives; select effective instructional strategies; and create attractive educational environments [44–47]. From the statements and points of view in Dimension 1: TITDA, we complied with 3 second-level items and 11 third-level measurement items, as shown in Table 3.

**Table 3.** Second and third-level items of TITDA and resource.

| Second-Level Items | Third-Level Measurement Items | Resource |
|---|---|---|
| The ability to analyze informatization students' situation | B1: I understand students' motivation and interests through technological resources. | Hudson and Ginns [48] |
| | B2: I grasp students' learning needs by technological resources. | Hudson and Ginns [48] |
| | B3: I gain students' knowledge bases by technological resources. | Hudson and Ginns [48] |
| The ability to select informatization teaching strategies | B4: I set teaching objectives by data analysis tools. | Papanastasiou and Angeli [49] |
| | B5: I combine with multimedia tools to select appropriate teaching methods. | Hsu [42] |
| | B6: I integrate multimedia tools into all kinds of teaching activities. | Papanastasiou and Angeli [49] |
| | B7: I use digital teaching resources to make teaching materials more clear. | Ku et al. [50] |
| | B8: I make courseware by common software and subject software. | Sailer et al. [51] |
| The ability to create informatization teaching situation | B9: I create immersive teaching environments by technological resources. | Akram and Zepeda [52] |
| | B10: I ensure multimedia tools and technological resources' normal usage in the classroom. | Sailer et al. [51] |
| | B11: I create technological environments conducive to the development of students' mental health. | Tondeur et al. [53] |

The third dimension is teachers' informatization teaching implementation ability (TITIA) which occurs during the lessons and relates to using technologies to achieve innovative instructions. Teaching implementation is the process of applying blueprints. TITIA encompasses teacher–student interaction; classroom organization and management;

and presents instructional content [54–56]. From the statements and points of view in Dimension 1: TITIA, we complied with 3 second-level items and 11 third-level measurement items, as shown in Table 4.

**Table 4.** Second and third-level items of TITIA and resource.

| Second-Level Items | Third-Level Measurement Items | Resource |
|---|---|---|
| The ability to teach in informatization classroom | C1: I use technological sources to achieve teaching innovation. | Zelkowski et al. [57] |
| | C2: I interact with students through teaching software and mobile devices. | Wei et al. [58] |
| | C3: I guide students to use mobile devices and communication software for group cooperation. | Ku et al. [50] |
| | C4: I guide students to use mobile devices and online learning platforms for autonomous learning. | Al Ansari et al. [59] |
| The ability to organize in informatization classroom | C5: I manage class teaching time properly with common software. | Pianta et al. [60] |
| | C6: I get students' feedbacks by technological resources and adjust the rhythm of the lessons quickly. | Sailer et al. [51] |
| | C7: I deal with emergencies caused by technical failures during lessons. | Martin and Sass [61] |
| | C8: I encourage all students to participate in informatization lessons actively. | Al Ansari et al. [59] |
| The ability to express in informatization classroom | C9: I express the contents and principles to students through common software and subject software. | Wen and Shih [41] |
| | C10: I carry out experiments and play videos through technological resources. | Ku et al. [50] |
| | C11: I construct mind maps and generalize knowledge structures by technological resources. | Hudson and Ginns [48] |

The fourth dimension is teachers' informatization teaching evaluation ability (TITEA), and it takes place during and after lessons. TITEA highlights that teachers make reasonable judgments about their teaching and students' learning by using technologies. TITEA includes designing a variety of evaluation methods; using different evaluation tools to assess students' learning situations and teachers' educating process; and providing constructive feedback [47,62,63]. From the statements and points of view in Dimension 1: TITEA, we complied three second-level items and nine third-level measurement items, as shown in Table 5.

**Table 5.** Second and third-level items of TITEA and resource.

| Second-Level Items | Third-Level Measurement Items | Resource |
|---|---|---|
| The ability to design informatization evaluation | D1: I develop personalized assessment schemes for each student by technological resources. | Akram and Zepeda [52] |
| | D2: I design diversified students' assessment methods through technological resources. | Tondeur et al. [53] |
| | D3: I set up teachers' evaluation, students' self-evaluation, students' mutual evaluation by technological resources. | Ay et al. [64] |

**Table 5.** *Cont.*

| Second-Level Items | Third-Level Measurement Items | Resource |
|---|---|---|
| The ability to application of informatization evaluation | D4: I create electronic portfolios for students to record their learning process. | Akram and Zepeda [52] |
| | D5: I find and solve students' learning problems accurately by data analysis tools. | Al Ansari et al. [59] |
| | D6: I conduct students' tests and exercises on mobile devices and network teaching platforms. | Jamieson-Proctor et al. [65] |
| The ability to form feedback of informatization evaluation | D7: I show evaluation results to colleagues, parents and students by multimedia tools. | DeLuca et al. [66] |
| | D8: I obtain students' learning feedbacks and adjust teaching plans by technological resources. | DeLuca et al. [66] |
| | D9: According to the evaluation results, I provide personalized coaching and tiered coaching for students by network teaching platforms. | Al Ansari et al. [59] |

## 4. Methods

### 4.1. Item Generation

As shown in the literature review section, current studies on TITA pay particular attention to teachers' informatization teaching metacognitive ability (TITMA), teachers' informatization teaching design ability (TITDA), teachers' informatization teaching implementation ability (TITIA) and teachers' informatization teaching evaluation ability (TITEA). According to the literature review, each domain includes several items. For example, Liu et al. [9] offered their explanation for TITDA in terms of the following four items: choose the knowledge points, select the media type, design appropriate teaching resources and effectively design each stage. This study adopted three steps to analyze and compare these detailed items: Firstly, we merged the same or similar items into one new item, and the items inconsistent with our definition were deleted [67]; secondly, we grouped all detailed items into three levels, which enabled the entire TITA scale to be more precise and accurate. Finally, we obtained potential TITA scale items for each aspect. Since all items were developed based on the literature published in Chinese, in order to ensure that the meanings of the TITA scale in the English version were in line with the Chinese version, the items were translated into English, and their legibility and readability were confirmed by two experts in the field with high language (i.e., English) abilities. These items were then translated back into Chinese to compare semantic differences between the two versions until there was no difference in the end.

To test whether the TITA instrument can be successfully applied to the local educational informatization context, we invited three experts who had more than 10 years of instructional experience and were familiar with local conditions of educational informatization. The experts were two primary and secondary school teachers and one researcher in the field of educational technology. The experts were from Nanjing and Suzhou cities, China. The TITA scale was revised according to their suggestions before conducting the questionnaire surveys. The final English TITA scale can be found in the Appendix A.

The TITA scale was administered to teachers. The five points Likert-type options were used to gain the teachers' tendencies, and they were labeled as strongly disagree, disagree, general, agree and strongly agree. A four-step weight calculation was carried out, and it included subjective weight calculation, objective weight calculation, combination weight calculation and weight normalization calculation. As many steps needed to be discussed, we carried out this research from January 2021 to October 2021. It lasted 10 months.

### 4.2. Participants

Snowball convenience sampling was used for recruiting teachers to participate in surveys, and the questionnaires were distributed through the Internet. The participants were Chinese primary and secondary school teachers from public schools. In the first stage of snowball convenience sampling, we selected 180 teachers from Nanjing city, Zhenjiang city and Wuxi city in Jiangsu province, China. There was no detailed demographic information at that stage. In the second stage of snowball convenience sampling, 223 teachers from Nanjing city, Zhenjiang city, Wuxi city and Suzhou city in Jiangsu province, China, were invited to complete the questionnaire. The first questionnaire survey was in March 2021; the second questionnaire survey was in August 2021. The time interval between the two surveys was six months. The participants were informed of the research purpose, research procedure, the definition of TITA, the meaning of each choice and the responsibility for this study before they filled out the questionnaire. The participants were told that the survey was completely anonymous. In total, 403 teachers participated in this survey. Demographic information from the second stage is reported in Table 6.

**Table 6.** Demographic information of participants in the second stage (*n* = 223).

| Characteristics | Items | Frequency | Percentage |
| --- | --- | --- | --- |
| Sex | Male | 74 | 33.18% |
| | Female | 149 | 66.82% |
| Teaching experience | More than 16 years | 14 | 6.28% |
| | 11–15 years | 28 | 12.56% |
| | 6–10 years | 72 | 32.29% |
| | Less than 5 years | 109 | 48.88% |
| School position | General (subject) teachers | 181 | 81.17% |
| | Principal | 12 | 5.38% |
| | Director | 30 | 13.45% |
| Education background | Ph.D. | 7 | 3.14% |
| | Master's degree | 107 | 47.98% |
| | Undergraduate | 104 | 46.64% |
| | Junior college | 5 | 2.24% |

### 4.3. Data Analysis

Wu [68] suggested that researchers can divide the entire scale into different subscales if the scale was determined based on theoretical exploration and was reviewed by the experts. To gain an accurate and precise scale for assessing TITA, we divided the whole scale into four subscales and then administered factor analysis for each subscale. There were four factors in TITMA that involved 17 items (A1–A17), three factors in TITDA that contained 15 items (B1–B15), three factors in TITIA that included 13 items (C1–C13) and three factors in TITEA that embraced 12 items (D1–D12).

Two sets of samples of EFA (*n* = 180) and CFA (*n* = 223) were used to test the instrument for assessing TITA. SPSS 25 was used to perform EFA and to compute Cronbach's alpha. The CFA was conducted in Amos 21.

EFA and reliability. The value of Kaiser–Meyer–Olkin (KMO) and Bartlett's test of sphericity was calculated to determine whether the data collected could be used for factor analysis [69]. We also applied the principal components analysis (PCA) to reveal the internal structure of multiple items and reduce the items of dimensions. Due to the complexity of ability, oblique rotation (direct oblimin rotation) was used to provide a more accurate solution for the TITA scale. Cronbach's alpha coefficient was used to measure the reliability of the entire TITA scale and each subscale.

CFA. CFA was used to test the convergent validity and goodness of model fit of the scale. If the factor loading of all items is greater than 0.5, it means the convergent validity of the model is good [44]. Model fit was evaluated using the following statistics: chi-square ($\chi^2$), chi-square statistic ($\chi^2/df$), root mean square error of approximation (RMSEA), standardized root means square residual (SRMR) and comparative fit index (CFI). According to Kline [70], Hu and Bentler [71] and Hooper et al. [72], a perfect model fit requires that $\chi^2$ need to be greater than 0.05, $\chi^2/df$ less than three, RMSEA and SRMR below 0.05, and TLI and CFI greater than 0.90.

Weight calculation. The weight of the TITA scale was calculated through Yaahp software for subjective weight calculation, CRITIC method for objective weight calculation, OLS method for combination weight calculation and Matlab software for weight normalization method.

## 5. Results

### 5.1. Exploratory Factor Analysis

The KMO and Bartlett's test of sphericity were conducted to test whether the samples collected could be suitable for EF. The KMO values of the four subscales of TITMA (0.914), TITDA (0.875), TITIA (0.861) and TITEA (0.829) were higher than 0.8; the results of Bartlett's test of sphericity reached the significant level ($p = 0.000 < 0.05$), which indicated that the samples collected were appropriate for EFA. Moreover, we did the KMO and Bartlett's test twice, and both of them met the requirements.

PCA was used twice in this study; the final result reported that there were four factors involved 12 items in TITMA (A1–A12), three factors covered 11 items in TITDA (B1–B11), three factors contained 11 items in TITIA (C1–C11) and factors involved 9 items in TITEA (D1–D9). Each item's factor loading of subscales was greater than 0.4. A total of 14 items in the TITA scale were deleted in PCA, including A3, A5, A8, A12 and A14 in TITMA; B6, B9, B2 and B13 in TITDA; C2 in TITIA; and D1 and D11 in TITEA.

### 5.2. Confirmatory Factor Analysis

Several fit indices of the subscales of TITA were calculated in CFA. $\chi^2$, $\chi^2/df$, RMSEA, CFI and SRMR in each subscale all met critical values (Table 7). This result suggests that the four subscales of TITA had a good fit with the 223 samples. Moreover, Table 8 shows the TITA scale of the 43 items and that the factor loading of all factors was significant ($<0.50$). This indicates that the four subscales of TITA have good validity.

**Table 7.** Summary of model fit of the TITMA, TITDA, TITIA and TITEA subscales.

| Fit Indexes | Criterion | TITMA | TITDA | TITIA | TITEA |
|:---:|:---:|:---:|:---:|:---:|:---:|
| $\chi^2$ | $p > 0.05$ | 0.571 | 0.492 | 0.053 | 0.671 |
| RMR | <0.05 | 0.019 | 0.030 | 0.026 | 0.021 |
| RMSEA | <0.08 | 0.043 | 0.070 | 0.072 | 0.069 |
| CFI | >0.90 | 0.0909 | 0.939 | 0.925 | 0.934 |
| $\chi^2/df$ | 1 < NC < 3 | 1.368 | 2.099 | 2.538 | 2.986 |

**Table 8.** Standardized estimates of confirmatory factor analysis.

| Dimension | Factor | Factor Loading | Residual Variance | Item | Factor Loading | Residual Variance |
|---|---|---|---|---|---|---|
| TITMA | Informatization teaching philosophy | 0.92 | 0.85 | A1 | 0.78 | 0.61 |
| | | | | A2 | 0.76 | 0.58 |
| | | | | A3 | 0.77 | 0.59 |
| | Informatization teaching skills | 0.93 | 0.86 | A4 | 0.74 | 0.55 |
| | | | | A5 | 0.79 | 0.62 |
| | | | | A6 | 0.76 | 0.58 |
| | Informatization pedagogical knowledge | 0.79 | 0.63 | A7 | 0.50 | 0.25 |
| | | | | A8 | 0.53 | 0.28 |
| | | | | A9 | 0.50 | 0.25 |
| | Informatization professional development | 0.88 | 0.78 | A10 | 0.77 | 0.60 |
| | | | | A11 | 0.75 | 0.57 |
| | | | | A12 | 0.79 | 0.63 |
| TITCA | The ability to analyze informatization students' situation | 0.89 | 0.80 | B1 | 0.69 | 0.47 |
| | | | | B2 | 0.69 | 0.47 |
| | | | | B3 | 0.69 | 0.48 |
| | The ability to select informatization teaching strategies | 0.91 | 0.82 | B4 | 0.67 | 0.45 |
| | | | | B5 | 0.68 | 0.46 |
| | | | | B6 | 0.71 | 0.50 |
| | | | | B7 | 0.69 | 0.48 |
| | | | | B8 | 0.69 | 0.47 |
| | The ability to create informatizationteaching situation | 0.91 | 0.84 | B9 | 0.69 | 0.47 |
| | | | | B10 | 0.70 | 0.40 |
| | | | | B11 | 0.69 | 0.48 |
| TITIA | The ability to teachin informatization classroom | 0.97 | 0.94 | C1 | 0.63 | 0.40 |
| | | | | C2 | 0.63 | 0.40 |
| | | | | C3 | 0.63 | 0.39 |
| | | | | C4 | 0.62 | 0.39 |
| | The ability to organize in informatization classroom | 0.90 | 0.81 | C5 | 0.62 | 0.39 |
| | | | | C6 | 0.67 | 0.45 |
| | | | | C7 | 0.68 | 0.46 |
| | | | | C8 | 0.66 | 0.44 |
| | The ability to express in informatization classroom | 0.92 | 0.85 | C9 | 0.66 | 0.44 |
| | | | | C10 | 0.65 | 0.42 |
| | | | | C11 | 0.68 | 0.46 |

**Table 8.** *Cont.*

| Dimension | Factor | Factor Loading | Residual Variance | Item | Factor Loading | Residual Variance |
|---|---|---|---|---|---|---|
| TITEA | The ability to design informatization evaluation | 0.91 | 0.83 | D1 | 0.68 | 0.46 |
| | | | | D2 | 0.70 | 0.49 |
| | | | | D3 | 0.67 | 0.44 |
| | The ability to application of informatization evaluation | 0.89 | 0.79 | D4 | 0.71 | 0.50 |
| | | | | D5 | 0.61 | 0.37 |
| | | | | D6 | 0.65 | 0.42 |
| | The ability to form feedback of informatization evaluation | 0.94 | 0.88 | D7 | 0.67 | 0.45 |
| | | | | D8 | 0.68 | 0.47 |
| | | | | D9 | 0.70 | 0.49 |

*5.3. Internal Consistency*

Cronbach's alpha value reflected the internal consistency reliability coefficient of four subscales. This study tested internal consistency twice. As seen in Table 9, Cronbach's alpha coefficients for four subscales ranged between 0.820 and 0.900. All Cronbach's alpha values of subscales were higher than 0.800. After EFA and CFA, the retest reliability coefficient of the TITA scale was calculated, and the value was 0.970. Coefficients for four subscales were found to be between 0.880 and 0.900. Therefore, we may conclude that the TTATT scale and seven factors are reliable.

**Table 9.** Reliability of the TITMA, TITDA, TITIA and TITEA subscales.

| Scale | Cronbach's Alpha 1 | Number of Items | Cronbach's Alpha 2 | Number of Items |
|---|---|---|---|---|
| TITMA | 0.898 | 17 | 0.900 | 12 |
| TITDA | 0.875 | 15 | 0.887 | 11 |
| TITIA | 0.844 | 13 | 0.895 | 11 |
| TITEA | 0.823 | 12 | 0.885 | 9 |
| TITA | | | 0.970 | 43 |

Cronbach's Alpha 1: before EFA and CFA; Cronbach's Alpha 2: after EFA and CFA.

*5.4. Weight Calculation*

The weight of the TITA scale was calculated through comprehensive values obtained from four steps. We reported the final result in Appendix A.

**6. Discussion**

As Aggarwal [73] mentioned, if educators could not use new technologies to innovate teaching activities, they would not be able to impart knowledge to students effectively. Comi et al. [74] speculated that the limitation of ICT might inhibit students' learning and prevent them from gaining critical and inquiry thinking skills. The above statements also can reveal such a logic of TITA and students' learning: TITA not only produces an influence on teachers' professional development but also has an impact on students' learning. Therefore, to help students master more knowledge and skills, teachers' teaching ability should be improved with supporting educational technologies. This study took notation to primary and secondary school teachers and developed a comprehensive instrument from the existent theories and models to assess TITA.

Several scholars developed various models to evaluate teachers' abilities and attitudes in the technological environment. Lu, Tsai and Wu [75] designed a scale to survey the infrastructure and application of ICT in middle and primary schools in urban areas and rural areas in China. To investigate teachers' attitudes about technology, Xu, Williams and

Gu [76] developed and validated the technology teachers' attitude toward the technology (TTATT) scale. Sang et al. [77] described the development and validation of the Chinese pre-service teachers' technological pedagogical content knowledge (CTPCK) scale. Based on the standards and a series of evaluation models, the TITA scale designed in this study involves teachers' abilities such as technology integration, design, implementation and evaluation abilities, etc. The TITA scale retains the advantages of the existing typical models and has its own characteristics. The design concept of the TITA measurement instrument is to conduct a more comprehensive assessment of teachers' teaching ability under the context of technology, which is similar to the research goals of Koehler et al. [22] and Schmid et al. [38].

It could be said that the items of the TITA instrument include teachers' informatization integration abilities based on the components of the teaching process. In this sense, the TITA framework has some differences from other studies focused on developing an elaborated model of the teaching-technology-related models, such as ICT and TPACK. The ICT–TPCK model developed by Angeli and Valanides [78] has a framework with five factors pedagogy, ICT, content, context and learners. This model is composed of teachers' comprehensive ability based on students' learning. Although TITA and ICT-PACK are student-centered learning, the TITA model differs from the ICT–TPCK framework. The TITA model focuses on students' learning situations in each link, highlighting that teaching, learning and technology are integrated rather than separate. Teachers must consider the three factors' relationship when they use technology. Another model based on the TPACK framework was developed by Cox and Graham [79]; this model provides subject-specific strategies and topic-specific strategies. Compared to Cox and Graham's model [79], TITA does not have a framework for specific topic content. That is, TITA is based on a generic teaching evaluation model in terms of an informatization teaching environment. Additionally, it differs from the technological pedagogical content knowledge scale in Yurdakul et al.'s study [80], whose four dimensions of TPACK were adapted from Koehler and Mishra's study [22] and consist of design, exertion, ethics and proficiency. Moreover, the TITA instrument also differs from TPACK.xs by Schmid et al. [38], including seven dimensions (e.g., TPK, PCK, CK) to assess pre-service teachers' teaching competencies with technology. Compared to Yurdakul et al. [80] and Schmid et al.'s [38] models, the TITA model pays more attention to teachers' teaching and students' learning; the whole instrument is developed based on the teaching process. Thus, it is considered that the TITA instrument can be distinguished from other relevant instruments and models in the literature.

The TITA instrument is different from the structures of the teacher's ability to apply educational technology (TAAET) series instruments [53,77,81] and the teacher's ability to apply information technology (TAAIT) sequence instruments [41,82]. The reason is that the purposes of TITA, TAAET and TAAIT are different. TAAET emphasizes that teachers use educational resources, optimize the educational process and integrate teaching links under the guidance of modern educational concepts and modern educational theories. TAAIT refers to teachers' ability to process, apply and deal with information technology to ensure the smooth progress of education and teaching activities. TITA highlights the teaching process, which emphasizes the comprehensive practical ability of teachers to integrate information technology with teaching design, teaching implementation, teaching evaluation and other links for the purpose of promoting students' development. From this perspective, it can be seen that TAAIT is the foundation of using technology to conduct the educational process and TAAET includes TITA. TITA is more related to teachers' instructional practices compared to TAAIT and TAAET. TITA refers to the ability of teachers to integrate technology with the whole teaching process to achieve the best teaching effect in the specific teaching process. Therefore, this study proposed a TITA structure that included four dimensions: TITMA, TITDA, TITIA and TITEA. In order to evaluate TITA more accurately, the TITA instrument was developed according to the educational process before, during and after lessons. In addition, this study also paid attention to teachers' metacognition. Metacognition is the process of thinking about one's own thinking and

learning [83]. In education, the metacognitive ability is very important because they help regulate own learning/teaching process. For that reason, TITMA is composed of teaching philosophy, teaching skills, pedagogical knowledge and professional development. Instead of designers' subjective judgment and development, dimensions, factors and items in the TITA scale adhered to the solid theoretical deduction. In addition to plenty of related authoritative literature, the standards of TAAET and TAAIT also were referenced in the TITA instrument.

Furthermore, compared with previous studies, diverse types of technology were manifested [6,25,26]. For example, common software, subject software, digital teaching resources, network teaching platforms, multimedia tools, mobile devices and online learning communities were included in this study. More researchers were accustomed to educational technology simply as a whole but ignored that different technology categories existed in instructional activities. These easily resulted in technology not being expressed clearly by researchers in the TITA instrument, thus forming a blurred assessment of TITA. Another enhancement is that the TITA was more meticulous of the TITA instrument. For instance, most studies only divided their instrument into two levels; this might bring out that the classification of the entire instrument could not be detailed enough and also caused items correlated with the topic to be not considered in the instrument. In order to avoid such issues, this study constructed the TITA instrument from three levels: four first dimensions (TITMA, TITDA, TITIA and TITEA), 13 second-level factors and 43 third-level measurement items. It provides a more scientific and precise instrument to evaluate TITA.

Teachers, researchers, and education leaders can readily use the TITA scale as a "compass" to take different strategies to improve teachers' ability in the informatization environment. From this scale, the shortcomings of teachers in the teaching process could be measured, and then any effective strategies to solve these issues could be used. For example, suppose teachers received low scores in innovatively applying diverse technologies in the educational process. In that case, some corresponding recommendations could be given to teachers to help them improve their teaching skills. Additionally, researchers and scholars can further explore the reasons for this situation based on the scores and then provide feedback for teachers and education leaders. Furthermore, the TITA scale may also offer policy decisions for education leaders. For instance, school leaders can propose multi-level training schemes for teachers with different abilities according to the results of TITA measurement.

Moreover, teachers, as instructional designers, using the instrument for assessing TITA can help other teachers think about how to interpret new knowledge to students better. For example, if teachers wanted to provide precision instructions for students' learning, they could use technological resources to achieve it. Before lessons, teachers distribute learning tasks through network teaching platforms and then design instructional links and content according to students' learning effects. During the lessons, teachers can adjust teaching rhythm and achieve precise teaching in the lessons by getting the results of students' learning responses through big data analysis. After the lessons, teachers can send personalized learning resources to students.

Using the instrument for assessing TITA in Chinese primary and secondary schools also can help students' development. In the TITA measurement instrument, we focus not only on teachers' teaching situations but also on students' learning situations in the informatization teaching environment. It is rarely mentioned in previous studies. Teachers play an essential role in promoting students' development in education. Teachers are the facilitators of students' personalized learning; they design great personalized learning strategies for students through the online platform. Teachers are evaluators of students' learning process. With the help of big data analysis technology, teachers collect and assess students' learning status comprehensively and give students feedback in time. Teachers are the creators of students' learning situations. In the era of artificial intelligence, teachers can reorganize teaching content based on real and virtual learning environments and create learning situations that adapt to different learning content. The changes in the teachers'

roles in the information era have put forward new requirements for teachers' teaching ability. Therefore, the TITA instrument designed the items, including guiding students' learning efficiently, e.g., B1, B2, B3, C3, D8, etc.

In addition, the development of the TITA scale in primary and secondary schools also provides the possibility for further research. The instrument developed in this study can support more high-quality empirical studies in the field of TITA. The TITA instrument affords educators and researchers a diagnostic tool that can be used to assess the current status and problems in teacher informatization teaching.

## 7. Conclusions

Improving TITA is crucial to promoting the sustainable development of education informatization. Teachers adopt diversified technologies in the teaching process to provide students with rich learning resources, effective learning tools, multiple learning approaches and lively learning environments. At the same time, technologies are used to enable more teachers to share sustainable teaching ideas and achievements and promote sustainable teaching actions. In this context, this study developed a measurement instrument to measure teachers' informatization teaching level, help improve the quality of teachers in the information age and promote education for sustainable development. In order to establish the theoretical framework of TITA, this study firstly reviewed the definitions of TITA. Secondly, four dimensions, 13 second-level factors and 43 third-level measurement items for assessing TITA were derived from the essential theories and models of the related literature. Finally, the instrument for evaluating TITA was validated by experts and tested by EFA, CFA and a reliability test. Thus, we may argue that this study constructed a more concrete, precise and valid TITA instrument.

A limitation of this study is that the TITA scale was developed by the actual status quo in the eastern cities of China, such as Nanjing, Suzhou and Zhenjiang. The majority of primary and secondary schools are equipped with iPads, computers, electronic whiteboards, etc., in the classroom in these cities, and the current instrument has the potential to be applicable to the tech-rich classroom. Therefore, whether the TITA scale can be used in different technological contexts and its universality needs further verification. In addition, although this study deduced four dimensions of the TITA scale from the influential literature, experts and quantitative analysis verified its content validity. However, this study lacks an investigation of the present situation of teachers' application of technology before developing the TITA scale. We believe that doing a pre-status survey could help researchers construct a more realistic scale. Therefore, if it is applied in others' studies, it is better to do a status survey to test whether the scale is appropriate for such context.

**Author Contributions:** Conceptualization, methodology, validation, S.Y., R.S. and R.Y.; formal analysis, S.Y.; investigation, S.Y.; resources, S.Y.; data curation, S.Y. and Y.L.; writing—original draft preparation, S.Y. and R.S.; writing—review and editing, S.Y. and R.S.; supervision, S.Y. and R.S. All authors have read and agreed to the published version of the manuscript.

**Funding:** This research received no external funding.

**Institutional Review Board Statement:** Not applicable.

**Informed Consent Statement:** The data for the study are not available publicly.

**Data Availability Statement:** Not applicable.

**Acknowledgments:** We would like to thank the teachers who voluntarily participated in this study.

**Conflicts of Interest:** The authors declare no conflict of interest in the publication of this research article.

# Appendix A

**Table A1.** TITA instrument in primary and secondary schools.

| First-Level Dimensions | Second-Level Items | Third-Level Measurement Items |
|---|---|---|
| TITMA (0.2960) | Informatization teaching philosophy (0.2768) | A1: TITA is a core component of teachers' competence. (0.2897) |
| | | A2: Technology is important for teaching. (0.3114) |
| | | A3: It is necessary to use technology to optimize teaching. (0.3989) |
| | Informatization teaching skills (0.2739) | A4: I explore technological resources for students' learning. (0.5052) |
| | | A5: I grasp how to use different multimedia tools. (0.2405) |
| | | A6: I have several methods to get digital instructional resources. (0.2543) |
| | Informatization pedagogical knowledge (0.2218) | A7: I master comprehensive knowledge of education. (0.3198) |
| | | A8: I have pedagogical knowledge. (0.3444) |
| | | A9: I possess computer ethics and information security knowledge. (3359) |
| | Informatization professional development (0.2274) | A10 I use online learning communities to improve myself. (0.3843) |
| | | A11: I communicate with experts by technological resources to improve myself. (0.2826) |
| | | A12: I solve teaching problems with common software tools. (0.3331) |
| TITDA (0.2407) | The ability to analyze informatization students' situation (0.3274) | B1: I understand students' motivation and interests through technological resources. (0.2828) |
| | | B2: I grasp students' learning needs by technological resources. (0.3222) |
| | | B3: I gain students' knowledge bases by technological resources. (0.3950) |
| | The ability to select informatization teaching strategies (0.4073) | B4: I set teaching objectives by data analysis tools. (0.1690) |
| | | B5: I combine with multimedia tools to select appropriate teaching methods. (0.1598) |
| | | B6: I integrate multimedia tools into all kinds of teaching activities. (0.2969) |
| | | B7: I use digital teaching resources to make teaching materials more clear. (0.1532) |
| | | B8: I make courseware by common software and subject software. (0.2212) |
| | The ability to create informatization teaching situation (0.2652) | B9: I create immersive teaching environments by technological resources. (0.3341) |
| | | B10: I ensure multimedia tools and technological resources' normal usage in the classroom. (0.3306) |
| | | B11: I create technological environments conducive to the development of students' mental health. (0.3353) |

**Table A1.** *Cont.*

| First-Level Dimensions | Second-Level Items | Third-Level Measurement Items |
|---|---|---|
| TITIA (0.3055) | The ability to teach in informatization classroom (0.4306) | C1: I use technological sources to achieve teaching innovation. (0.2340) |
| | | C2: I interact with students through teaching software and mobile devices. (0.2145) |
| | | C3: I guide students to use mobile devices and communication software for group cooperation. (0.2712) |
| | | C4: I guide students to use mobile devices and online learning platforms for autonomous learning. (0.2804) |
| | The ability to organize in informatization classroom (0.3206) | C5: I manage class teaching time properly with common software. (0.1887) |
| | | C6: I get students' feedbacks by technological resources and adjust the rhythm of the lessons quickly. (0.2799) |
| | | C7: I deal with emergencies caused by technical failures during lessons. (0.1715) |
| | | C8: I encourage all students to participate in informatization lessons actively. (0.1715) |
| | The ability to express in informatization classroom (0.2489) | C9: I express the contents and principles to students through common software and subject software. (0.4565) |
| | | C10: I carry out experiments and play videos through technological resources.(0.2558) |
| | | C11: I construct mind maps and generalize knowledge structures by technological resources. (0.2877) |
| TITEA (0.1578) | The ability to design informatization evaluation (0.3177) | D1: I develop personalized assessment schemes for each student by technological resources. (0.2869) |
| | | D2: I design diversified students' assessment methods through technological resources. (0.3591) |
| | | D3: I set up teachers' evaluation, students' self-evaluation, students' mutual evaluation by technological resources. (0.2540) |
| | The ability to application of informatization evaluation (0.3177) | D4: I create electronic portfolios for students to record their learning process. (0.2669) |
| | | D5: I find and solve students' learning problems accurately by data analysis tools. (0.3611) |
| | | D6: I conduct students' tests and exercises on mobile devices and network teaching platforms. (0.3720) |
| | The ability to form feedback of informatization evaluation (0.3266) | D7: I show evaluation results to colleagues, parents and students by multimedia tools. (0.3353) |
| | | D8: I obtain students' learning feedbacks and adjust teaching plans by technological resources. (0.3008) |
| | | D9: According to the evaluation results, I provide personalized coaching and tiered coaching for students by network teaching platforms. (0.3639) |

There are operational definitions of terminologies (Wu, 2014) [23].

1. Technological resources are all technologies towards education: common software, subject software, digital teaching resources, network teaching platforms, multimedia tools, mobile devices and online learning communities etc.
2. Common software is used widely in different teaching activities: office software and voice communication software, etc.

3. Subject software refers to teaching software suitable for different subjects: mathematical geometry sketchpad, virtual chemistry laboratory, English listening software, etc.
4. Digital teaching resources refer to all resources related to education: teaching materials, network courseware, e-books, theme websites, etc.
5. Network teaching platforms refer to online platforms that promote teaching activities: resource platforms, interactive platforms, evaluation platforms, etc.
6. Multimedia tools refer to hardware equipment that promotes teaching activities: computer, electronic whiteboard, etc.
7. Mobile devices are portable learning tools: laptop, iPad, mobile phone.
8. Online learning communities is online platforms that support teachers in learning, communication, seminars, etc.

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
