# Peer review of "Developing and Validating an Instrument for Measuring Teachers’ Informatization Teaching Ability in Primary and Secondary Schools in China for the Sustainable Development of Education Informatization"

_sustainability, doi:10.3390/su14116474_

Round 1

Reviewer 1 Report

The paper presents the results of the study: Developing and Validating an Instrument for Measuring Teachers’ Informatization Teaching Ability in Primary and Secondary Schools in China for the Sustainable Development of Education

The scientific contribution is rather low. I am wondering why not established  and well-know measurement instruments such as TPACK are considered - why another instrument? (please compare: Developing a short assessment instrument for Technological Pedagogical Content Knowledge (TPACK.xs) and comparing the factor structure of an integrative and a transformative model - ScienceDirect)

The authors write at the end: "The TITA instrument is different from the structures of the teacher’s ability to apply educational technology (TAAET) series instruments [54, 55] and the teacher’s ability apply information  technology (TAAIT) sequence instruments -> but why is it different? The arguments are missing, the rationale of the research goals are not convincing

Furthermore, the new instrument is used as a self-assessment tool for reflection - it is not a bad thing, however, many empirical studies discuss the problems with self-assessments. As we already have such self-assessment instruments it is really not understandable why there is a research gap for another one.

Relevant literature is not considered, Tondeur is mentioned (2016) but not newer literature, e.g. What to teach? Strategies for developing digital competency in preservice teacher training - ScienceDirect

Author Response

Dear Reviewer 1,

We highly appreciate your time to read our manuscript and provide your feedback. Thank you for your valuable and critical comments concerning our manuscript. We have studied all your comments carefully and have made relevant corrections in the manuscript. Please see our detailed responses in the attached file.

Sincerely, Authors

Reviewer 2 Report

The article "Developing and Validating an Instrument for Measuring Teachers’ Informatization Teaching Ability in Primary and Secondary Schools in China for the Sustainable Development of Education Informatization" deals with an important topic, especially in the field of educational sciences. Based on a literature review, the authors define TITA (Teachers' Informatization Teaching Ability), develop a theoretical framework for the TITA scale, test and validate the instrument. The paper is clearly written and follows the usual scientific structure. The title and the abstract correspond to the content of the paper. The results are well described and clear. The discussion of the results and the conclusions of the study are well developed.

I believe that a few points should be clarified before publication. Namely, the main one is methodological - the paper does not indicate when the research was carried out, what the time interval was between the first and second stage of teacher surveys. Another issue is related to the ethical aspect of the research - whether the consent of the study participants was obtained?

Author Response

Dear Reviewer 2,

Thank you for your valuable and very helpful comments to us. We have studied them thoroughly and made relevant modifications in the manuscript. Please find our detailed responses in the attached file. Please let us know if there is anything else we need to modify.

Sincerely, authors

Reviewer 3 Report

An indirect definition of ‘sustainable’ ought to be provided in the introduction.

It would be necessary to refer to the specific theory on the basis of which the TITA model was proposed. On the basis of what theoretical assumptions do the authors propose to analyze TITA according to selected dimensions (TITMA, TITDA, TITIA, TITEA)?

The process of selecting items for the study instrument is not clearly explained. it should be more precisely explained what initial tools of each dimension were taken into account when developing the study instrument (authors, years).

The study would look more representative and convincing if its results were compared with those recently carried out outside of Chine. Therefore, analysis of the recent study results regarding the study subject (TITE) in other countries is recommended. The research tools which were used in the study look to be regionally scaled and have to be justified with the previous standardized tools for TITE.

Assessment of the results of the study in relation to other literature is unclear and not well organized in the discussion section.

Author Response

Dear Reviewer 3,

Thank you for your valuable and very helpful comments and suggestions. We have tied our best to address each of them. Please find our detailed responses in the attached file and let us know if there is anything else we need to do with the manuscript.

Sincerely, authors

Round 2

Reviewer 1 Report

Excellent Revision, my reviewer comments are completely considered 

Author Response

Dear Reviewer 1,

Thank you for your letter and the comments concerning our manuscript. It was your comments and suggestions that helped us improve the quality of this paper. Thank you very much!

Sincerely, Authors

Reviewer 3 Report

The authors of the definition of ‘sustainability’ should be given

Author Response

Dear Reviewer 3,

Thank you for your comment concerning our manuscript. This comment is valuable and helpful. Following your suggestion, we have added missing authors. Revisions in the text are shown using red highlights. We would love to thank you for allowing us to resubmit a revised copy of the manuscript and we highly appreciate your time and consideration.

Sincerely, Authors

Comment 1. The authors of the definition of ‘sustainability’ should be given.

Response 1. Thanks for the reviewer’s suggestion. We are sorry that we ignored this issue before. And we think it is necessary to give the definition of ‘sustainability’ and add its reference, which makes the contents more persuasive. Therefore, we added the three references in the corresponding place. The specific location is in the introduction section, page 1, lines 30 and 31. The added references are as follows:

  1. Scoones, I. Sustainability. Development in practice.2007, 17, 589-596.
  2. Huckle J and Sterling S (Eds). Education for sustainability. London, UK: Earthscan Publications Ltd.1996.
  3. Ruiz-Mallén, I., & Heras, M. What sustainability? Higher education institutions’ pathways to reach the Agenda 2030 goals. Sustainability. 2020, 12, 1290.